# Structural Kernel Search via Bayesian Optimization and Symbolical Optimal Transport

**Matthias Bitzer**    **Mona Meister**    **Christoph Zimmer**
Bosch Center for Artificial Intelligence, Renningen, Germany
{matthias.bitzer3,mona.meister,christoph.zimmer}@de.bosch.com

## Abstract

Despite recent advances in automated machine learning, model selection is still a complex and computationally intensive process. For Gaussian processes (GPs), selecting the kernel is a crucial task, often done manually by the expert. Additionally, evaluating the model selection criteria for Gaussian processes typically scales cubically in the sample size, rendering kernel search particularly computationally expensive. We propose a novel, efficient search method through a general, structured kernel space. Previous methods solved this task via Bayesian optimization and relied on measuring the distance between GP's directly in function space to construct a kernel-kernel. We present an alternative approach by defining a kernel-kernel over the symbolic representation of the statistical hypothesis that is associated with a kernel. We empirically show that this leads to a computationally more efficient way of searching through a discrete kernel space.

## 1 Introduction

In many real-work applications of machine learning, tuning the hyperparameters or selecting the machine learning method itself is a crucial part of the workflow. It is often done by experts and data-scientist. However, the number of possible methods and models is constantly growing, and it is becoming increasingly important to automatically select the right model for the task at hand. Bayesian optimization (BO) is a prominent method that can be used for model selection and hyperparameter tuning. It can handle black-box oracles with expensive function evaluations, which are two characteristics often encountered when doing model selection [17]. Important applications in this context are choosing the hyperparameters in the training process of deep neural networks (DNN) [17], or dealing with discrete and structured problems like choosing the architecture of DNN's [5, 11].

Gaussian processes (GP) are another important model-class. They are often utilized as surrogate models in BO [17], for time-series and statistical modeling [7, 3] or in active learning loops [23, 15]. The properties of GP's are mainly governed by its kernel that specifies the assumptions made on the underlying function. Choosing the right kernel is therefore a crucial part of applying GP's and is often done by the expert. Recent work [9] treated the kernel selection as a black-box optimization problem and used Bayesian optimization to solve it. This allowed searching over a highly structured, discrete space of kernels. However, their proposed kernel-kernel measures the distance between two GP's directly in function space, which is a computationally expensive task itself. This makes the method difficult to apply for the frequent scenarios where the evaluation of the model selection criteria requires only a medium amount of time.

We propose measuring the distance between two kernels via their symbolical representation of their associated statistical hypothesis. We utilize the highly general kernel-grammar, presented in [3], as underlying kernel space, where each kernel is build from base kernels and operators, like e.g.

$$\text{LIN} + ((\text{SE} \times \text{PER}) + \text{SE})$$

36th Conference on Neural Information Processing Systems (NeurIPS 2022).

forming effectively a description of the statistical hypothesis that is modeled by the GP. Our main idea is to build a distance over these symbolical descriptions, rather than measuring the distance between two GP's directly in function space. We employ optimal transport principles, known from neural-architecture search (see [11],[5]), to build a pseudo-distance between two hypotheses descriptions and use it to construct a kernel-kernel, which is subsequently utilized in the BO loop. We will show that the induced kernel search method is more efficient, in terms of number of function evaluations and computational time, compared to alternative kernel search methods over discrete search spaces.

The main challenge we encountered is the quantification of dissimilarity between two symbolical representations of kernels. We use the tree representation of each symbolical description and apply optimal transport distances over tree features. Subsequently, we empirically show that the deduced GP over GP's provides a well-behaved meta-model and show its advantages for kernel search. In summary, our contributions are:

1. We construct a pseudo-distance over GP's that acts over the symbolical representations of the underlying statistical hypothesis.

2. We use the pseudo-distance to construct a novel "kernel-kernel" and build it in a BO loop to do model selection for GP's.

3. We empirically show that our meta-GP model is well-behaved and that we outperform previous methods in kernel search over discrete kernel spaces.

**Related Work:** There is considerable existing work on constructing flexible kernels and learning their hyperparameters [1, 20, 21, 18, 22]. In these kinds of works, the structural form of the kernel is predefined and the free parameters of the kernel are optimized, often via marginal likelihood maximization. While being able to efficiently finding the hyperparameter due to the differentiability of the marginal likelihood, one still needs to predefine the structural form of the kernel in the first place. This is a hard task as one need to decide if long-range correlation or nonstationarity should be considered or if dimensions are ignored or not. Our method is build to automatically select the structure of the kernel. Some of the mentioned methods [20, 21] are able to approximate any stationary kernel via Bochner's Theorem and therefore consider a broad kernel space themselves. However, even elementary statistical hypotheses require nonstationary kernels, such as linear trends modeled by the linear kernel. Our search space is not restricted to stationary kernels.

We consider a highly general, discrete kernel space that is induced by the kernel grammar [3]. Recent work [9] used BO to search through this space via a kernel that measures similarity in function space. We also utilize BO but employ a fundamentally different principle of measuring the distance, which is computationally more efficient. We dedicate Section 3.2 to a more precise comparison to the method of [9]. Additionally, the original kernel grammar paper [3] suggested greedy search for searching through the kernel grammar. Furthermore, [4] employed a genetic algorithm based on cross-over mutations. We empirically compare against all approaches.

In the area of BO over structured spaces, our method is most similar to the neural-architecture search (NAS) procedures presented in [11, 5] who use optimal transport distances over features extracted from the graph-representation of neural networks. Compared to these methods we use OT principles to do model selection for GP's which is a fundamentally different task.

## 2 Background and Set-up

Our main task is efficient model selection for Gaussian processes. In order to provide background information, we give a small introduction to GP's and model selection for GP's. Subsequently, we present a review of the kernel-grammar [3] and show how we use it for our approach. We show how a kernel can be represented via a symbolical description. In Section 3, we will present how we use the symbolical description to construct a kernel over kernels and how we use it in the BO loop.

### 2.1 Gaussian Processes

A Gaussian process is a distribution over functions $f : \mathcal{X} \to \mathbb{R}$ over a given input space $\mathcal{X}$ which is fully characterized via the covariance/kernel function $k(x, x') = \mathbf{Cov}(f(x), f(x'))$ and the mean function $\mu(x) := \mathbb{E}[f(x)]$. We therefore can write as shorthand notation $f \sim \mathcal{GP}(\mu(\cdot), k(\cdot, \cdot))$. The

kernel can be interpreted as a similarity measure between two elements of the input space, with the GP assigning higher correlations to function values whose inputs are more similar according to the kernel. Furthermore, the kernel governs the main assumptions on the modeled function such as smoothness, periodicity or long-range correlations and therefore provides the inductive-bias of the GP. An important property of GP's is that they are not restricted to euclidean input spaces $\mathcal{X} \subset \mathbb{R}^d$, but can also be defined on highly structured spaces like trees and graphs, a property we will later use to define a GP over GP's.

While our method might also be used for model selection in classification, we consider from now on Gaussian processes regression. For regression, a dataset $\mathcal{D} = (\mathbf{X}, \mathbf{y})$ with $\mathbf{X} = \{x_1, \ldots, x_N\} \subset \mathcal{X}$ and $\mathbf{y} = (y_1, \ldots, y_N)^\mathsf{T} \in \mathbb{R}^N$ is given, where we suppose that $f \sim \mathcal{GP}(\mu(\cdot), k(\cdot, \cdot))$ and $y_i = f(x_i) + \epsilon_i$ with $\epsilon_i \overset{i.i.d}{\sim} \mathcal{N}(0, \sigma^2)$. Given the observed data $\mathcal{D}$ the posterior distribution $f|\mathcal{D}$ is again a GP with mean and covariance functions

$$\mu_\mathcal{D}(x) = \mu(x) + \mathbf{k}(x)^\mathsf{T}(\mathbf{K} + \sigma^2 I)^{-1}(\mathbf{y} - \mu(\mathbf{X})),$$
$$k_\mathcal{D}(x, y) = k(x, y) - \mathbf{k}(x)^\mathsf{T}(\mathbf{K} + \sigma^2 I)^{-1}\mathbf{k}(y)$$

with $\mathbf{K} = [k(x_m, x_l)]_{m,l=1}^N$ and $\mathbf{k}(x) = [k(x, x_1), \ldots, k(x, x_N)]^\mathsf{T}$ (see [13]). Probabilistic predictions can be done via the resulting predictive distribution $p(f^*|x^*, \mathcal{D}) = \mathcal{N}(\mu_\mathcal{D}(x^*), k_\mathcal{D}(x^*, x^*))$.

## 2.2  Model selection for GP's

Typically, the kernel $k_\theta$ comes with a set of parameters $\theta$ that can be learned via maximization of the marginal likelihood $p(\mathbf{y}|\mathbf{X}, \theta, \sigma^2) = \mathcal{N}(\mathbf{y}; \mu(\mathbf{X}), k_\theta(\mathbf{X}, \mathbf{X}) + \sigma^2\mathbf{I})$ or via maximum a posteriori (MAP) estimation, in case the parameters $\theta$ are equipped with a prior $p(\theta)$. This procedure is sometimes also called model selection, as one selects the hyperparameters of the kernel given the data (see [13]). However, we consider selecting the *structural form* of the kernel itself. The structural form of the kernel determines the statistical hypothesis that is assumed to be true for the data-generating process. Intuitively, the kernel is similar to the architecture in deep neural networks, which induces an inductive bias.

Our goal is to do model selection over a discrete, possibly infinite space of kernels $\mathbb{K} := \{k_1, k_2, \ldots\}$. As each kernel comes with its own parameters, we are actually dealing with a space of kernel families. Thus, when mentioning a kernel $k$ we associate it with its whole family over parameters $\{k_\theta|\theta \in \Theta\}$. Once a kernel is selected, predictions are done with learned kernel parameters (that usually are a by-product of calculating the model selection criteria). The parameters $\theta$ are potentially equipped with a prior $p(\theta)$ depending on the selection criteria. As mean function, we always utilize the zero mean function $\mu(x) := 0$ in case $\mathbf{y}$ is centered and a constant mean function otherwise. This is a common choice in GP regression. Given some model selection criteria $g : \mathbb{K} \to \mathbb{R}$ our task is solving $k^* = \arg\max_{k \in \mathbb{K}} g(k|\mathcal{D})$. While our method is not restricted to a specific model selection criteria, we focus on the model evidence $p(\mathbf{y}|\mathbf{X}, k)$, which is a well-known selection criteria for probabilistic models (see [9, 14, 8]). Given a prior on the kernel parameters $p(\theta)$ and the likelihood variance $p(\sigma^2)$, the log-model evidence of the marginalized GP is given as

$$g(k|\mathcal{D}) = \log p(\mathbf{y}|\mathbf{X}, k) = \log \int p(\mathbf{y}|\mathbf{X}, \theta, \sigma^2, k) p(\sigma^2) p(\theta|k) d\theta d\sigma^2.$$

This quantity can be approximated via Laplace approximation of $p(\theta, \sigma^2|\mathcal{D})$ (see Appendix A for details). Computing this approximation includes performing a MAP estimation of the GP parameters. Thus, once the log evidence has been computed, learned kernel hyperparameters $\theta_{\text{MAP}}$ are automatically provided. Performing the MAP estimation scales cubically in the data set size $N$, which renders model selection for GP's computationally intense.

## 2.3  Kernel Grammar

The kernel grammar introduced by [3] specifies a highly general search space over kernels. The grammar is based on the observation that kernels are closed under addition and multiplication, meaning, for kernels $k_1(x, x')$ and $k_2(x, x')$ also $k_1(x, x') + k_2(x, x')$ and $k_1(x, x') \times k_2(x, x')$ are kernels. Given some base kernels, such as the squared exponential kernel SE, the linear kernel LIN, the periodic kernel PER or the rational quadratic kernel RQ, different statistical hypotheses can be

generated via addition and multiplication. For example,

$$\text{LIN} + \text{PER} \times \text{SE}$$

describes a linear trend with a locally periodic component, such that it might be a useful hypothesis for time-series applications. For multidimensional data the base kernels are applied to single dimensions, denoted e.g. with $\text{SE}_i$ for the squared exponential kernel defined on dimension $i$. The grammar contains many popular hypotheses such as the ARD-RBF kernel that can be expressed via multiplication over the dimensions $\prod_{i=1}^{d} \text{SE}_i$ as well as additive models $\sum_{i=1}^{d} \text{SE}_i$ or polynomials of order $m$ with $\prod_{j=1}^{m} \text{LIN}$.

The specification of the complete search space is given as a grammar, where a base kernel is denoted as $\mathcal{B}$ and a subexpression is denoted with $\mathcal{S}$. For example, in the expression $\text{LIN} + (\text{PER} \times \text{SE})$ the expression $\text{PER} \times \text{SE}$ is a subexpression. Starting with all base kernels, the search space is defined as all kernels that can be reached via the following operations:

1. Add a base kernel to a subexpression: $\mathcal{S} \to \mathcal{S} + \mathcal{B}$
2. Multiply a subexpression with a base kernel: $\mathcal{S} \to \mathcal{S} \times \mathcal{B}$
3. Exchange a base kernel with another base kernel: $\mathcal{B} \to \mathcal{B}'$

Starting from the set of base kernels, these operations can lead to all algebraic expressions, showing the expressiveness of the grammar [3]. The authors of [3] suggest using greedy search to search through the grammar, where the kernel with the highest value for the selection criteria is selected and expanded with all possible operations. After the selection criteria is calculated on the neighbors, the search progresses to the next stage by expanding the neighbors of the best kernel found.

We consider a generalized notion of the kernel grammar, where we consider a set of base kernels $\{\mathcal{B}_1, \ldots, \mathcal{B}_r\}, r \in \mathbb{N}$ and a set of operators $\{T_1, \ldots, T_l\}, l \in \mathbb{N}$ where $T_j : \mathcal{K} \times \mathcal{K} \to \mathcal{K}, j = 1, \ldots, l$ are closed operators on the space of all kernel functions $\mathcal{K}$. Examples are addition and multiplication, but also the change-point operator which was considered in [7]. Thus, in general the grammar operations are

1. Apply operator $T_j$ onto a subexpression and a base kernel[1]: $\mathcal{S} \to T_j(\mathcal{S}, \mathcal{B})$
2. Exchange a base kernel with another base kernel: $\mathcal{B} \to \mathcal{B}'$

The considered kernel space can be defined precisely in the following way:

**Definition 1** *For $r, l \in \mathbb{N}$, let $\{k_1, \ldots, k_r\}$ be a set of (base-) kernels (symbollically represented as $\{\mathcal{B}_1, \ldots, \mathcal{B}_r\}$) and $\{T_1, \ldots, T_l\}$ a set of operators with $T_j : \mathcal{K} \times \mathcal{K} \to \mathcal{K}, j = 1, \ldots, l$. Let*

$$L_0 := \{k_1, \ldots, k_r\}$$
$$L_i := \{T_j(k_1, k_2) \mid k_1, k_2 \in L_{i-1}, j = 1, \ldots, l\} \cup L_{i-1}$$

*for $i = 1, \ldots, M$. We call $\tilde{\mathbb{K}} := L_M$ the kernel-grammar generated kernel space with depth $M$.*

### 2.4 Representation of Kernels

The base kernels and their combination via the operators describe the structural assumptions of the final kernel and imply the assumptions that are made in function space. Our main hypothesis is that the symbolical representation of the kernel, given by the subexpressions $\mathcal{S}$ and base kernels, already contains sufficient information for model selection. Therefore, we will define a kernel-kernel over the symbolical representations and utilize it for Bayesian optimization. Concretely, each kernel $k \in \tilde{\mathbb{K}}$ in our described space can be written as a tree $\mathcal{T}$, for example:

$$\text{LIN} + ((\text{SE} + \text{PER}) \times \text{SE}) \quad \longleftrightarrow$$

---

[1]In case the operator $T_j$ is not symmetric we also add the operation $\mathcal{S} \to T_j(\mathcal{B}, \mathcal{S})$. However, all operators considered in this work are symmetric.

Each operator $T_j$ and each base kernel $\mathcal{B}_i$ is represented by their respective name, where operators are the nodes of the tree and base kernels are the leafs. The way how the operators and base kernels are connected is represented through the tree structure. For a given expression tree $\mathcal{T}$, we denote the multiset of all subexpressions/subtrees as $\mathrm{Subtrees}(\mathcal{T})$. Furthermore, we consider paths to the leafs of the tree, for example, for the tree above one path to a leaf would be:

$$\mathbf{ADD} \longrightarrow \mathbf{MULT} \longrightarrow \mathbf{ADD} \longrightarrow \mathrm{PER}$$

We denote the multiset of all paths in the tree $\mathcal{T}$ as $\mathrm{Paths}(\mathcal{T})$. Lastly, we also consider the multiset of all base kernels that exist in a given expression tree as $\mathrm{Base}(\mathcal{T})$. For each described multiset, we denote the number of occurrences of element $\mathcal{E}$ in the multiset as $n(\mathcal{E})$. When building the multisets we also account for two symmetries in the elements that can be applied if an operator is associative and commutative, which we elaborate further in Appendix A.

Depending on the operators that are used, several trees can describe the same kernel $k \in \tilde{\mathbb{K}}$. For technical reasons, we denote with $f : \tilde{\mathbb{K}} \mapsto \mathcal{T}(\tilde{\mathbb{K}})$ a mapping that maps a given kernel $k \in \tilde{\mathbb{K}}$ to one tree $\mathcal{T}$ that induces this kernel, where $\mathcal{T}(\tilde{\mathbb{K}})$ denotes the set of all expression trees that can generate a kernel in $\tilde{\mathbb{K}}$ (see details in Appendix A).

## 3  Kernel-Kernel

Our kernel-kernel will be defined via a pseudo-metric over the expression trees. Optimal transport (OT) principles have proven themselves to be effective in BO methods over structured spaces, as shown in [5] and [11] for neural architecture search or in [6] for BO over molecule structures. We follow this line of work and also rely on optimal transport, although we only use a simple ground metric to allow for closed-form computations. To allow OT metrics to be used, we summarize each expression tree $\mathcal{T}$ to discrete probability distributions of their building blocks $\mathrm{Base}(\mathcal{T})$, $\mathrm{Paths}(\mathcal{T})$ and $\mathrm{Subtrees}(\mathcal{T})$ via

$$\omega_{\mathrm{base}} := \sum_{\mathcal{E} \in \mathrm{Base}(\mathcal{T})} \omega_{\mathcal{E}} \delta_{\mathcal{E}} \ , \qquad \omega_{\mathrm{paths}} := \sum_{\mathcal{E} \in \mathrm{Paths}(\mathcal{T})} \omega_{\mathcal{E}} \delta_{\mathcal{E}} \ , \qquad \omega_{\mathrm{subtrees}} := \sum_{\mathcal{E} \in \mathrm{Subtree}(\mathcal{T})} \omega_{\mathcal{E}} \delta_{\mathcal{E}} \ ,$$

where $\delta_{\mathcal{E}}$ is the Dirac delta and $\omega_{\mathcal{E}}$ is the frequency of the element $\mathcal{E}$ in the respective multiset. For example, the frequency of expression $\mathrm{SE} \times \mathrm{PER}$ in the multiset $\mathrm{Subtree}(\mathcal{T})$ is calculated as

$$\omega_{\mathrm{SE} \times \mathrm{PER}} = \frac{n(\mathrm{SE} \times \mathrm{PER})}{|\mathrm{Subtree}(\mathcal{T})|}.$$

Each probability distribution represents a different modeling aspect that is induced by a kernel $k \in \tilde{\mathbb{K}}$ and its corresponding expression tree $\mathcal{T}$:

1. $\omega_{\mathrm{base}}$ specifies *which* base kernels are present in $\mathcal{T}$, thus, which base assumptions in function space are included such as periodicity, linearity, local smoothness.

2. $\omega_{\mathrm{paths}}$ specifies *how* the base kernels in $\mathcal{T}$ are used, e.g. whether a periodic component is applied additively or multiplicatively.

3. $\omega_{\mathrm{subtrees}}$ specifies the *interaction* between the base kernels in $\mathcal{T}$, for example if $\mathcal{T}$ contains an addition of a linear and a periodic component or not.

Our pseudo-metric between kernels $k_1$ and $k_2$ (and its associated trees $\mathcal{T}_1$ and $\mathcal{T}_2$) uses all three modeling aspects via the optimal transport distances between the respective discrete probability distributions $\omega_{\mathrm{base}}$, $\omega_{\mathrm{paths}}$ and $\omega_{\mathrm{subtrees}}$.

In general the OT distance with ground metric $\tilde{d}$ between two discrete probability distribution $\omega_1 = \sum_{\mathcal{E} \in \Omega} \omega_{1,\mathcal{E}} \delta_{\mathcal{E}}$ and $\omega_2 = \sum_{\mathcal{E} \in \Omega} \omega_{2,\mathcal{E}} \delta_{\mathcal{E}}$ over $\Omega$ is defined as

$$W_{\tilde{d}}(\omega_1, \omega_2) = \inf_{\pi \in \mathcal{R}(\omega_1, \omega_2)} \int_{\Omega \times \Omega} \tilde{d}(\mathcal{E}, \mathcal{E}') \pi(d\mathcal{E}, d\mathcal{E}'), \tag{1}$$

where $\pi \in \mathcal{R}(\omega_1, \omega_2)$ is a combined probability distribution over $\Omega \times \Omega$ with marginal distribution $\pi(A \times \Omega) = \omega_1(A)$ and $\pi(\Omega \times B) = \omega_2(B)$ for Borel sets $A, B$. While for general ground metric

$\tilde{d} : \Omega \times \Omega \to \mathbb{R}$ the optimization problem in (1) can not be solved in closed-form we use as ground metric $\tilde{d}(\mathcal{E}, \mathcal{E}') = \mathbf{1}_{\mathcal{E} \neq \mathcal{E}'}$ which has as closed-form solution the total variation distance between $\omega_1$ and $\omega_2$ (see [19], p. 22), i.e.

$$W_{\tilde{d}}(\omega_1, \omega_2) = \frac{1}{2} \sum_{\mathcal{E} \in \Omega} |\omega_{1,\mathcal{E}} - \omega_{2,\mathcal{E}}|.$$

Utilizing this distance over the modeling assumptions $\omega_{\text{base}}$, $\omega_{\text{paths}}$ and $\omega_{\text{subtrees}}$ allows for fast computation of the final pseudo-metric and for a proper positive semi definite (p.s.d) kernel-kernel in the end [see Appendix C]. This is not guaranteed for general OT distances (see [11]). We define the final distance between two kernels $k_1$ and $k_2$ (and its associated trees $\mathcal{T}_1$ and $\mathcal{T}_2$) as a sum over the OT distances of their respective modeling components

$$
\begin{aligned}
d(\mathcal{T}_1, \mathcal{T}_2) := & \alpha_1 W_{\tilde{d}}(\omega_{1,\text{base}}, \omega_{2,\text{base}}) \\
& + \alpha_2 W_{\tilde{d}}(\omega_{1,\text{paths}}, \omega_{2,\text{paths}}) \\
& + \alpha_3 W_{\tilde{d}}(\omega_{1,\text{subtrees}}, \omega_{2,\text{subtrees}}),
\end{aligned}
\tag{2}
$$

where $\alpha_i \geq 0$ and $\sum_i \alpha_i = 1$ are weighting parameters that will later be learned automatically via marginal likelihood maximization for GPs.

In case the kernel grammar contains base kernels that act on single dimensions such as $\mathrm{SE}_i$, we define for each dimension $i = 1, \ldots, D$ an individual distribution over base kernels

$$\omega_{\text{base}}^{(i)} := \sum_{\mathcal{E} \in \text{Base}(\mathcal{T}, i)} \omega_{\mathcal{E}} \delta_{\mathcal{E}} \ ,$$

where $\text{Base}(\mathcal{T}, i)$ is the multiset of all base kernels in $\mathcal{T}$ defined on dimension $i$. We also include the empty-expression $\mathcal{E}_{\text{NULL}}$ for which $\omega_{\mathcal{E}_{\text{NULL}}} = 1$ whenever no base kernel of dimension $i$ is contained in $\mathcal{T}$. $\omega_{\text{base}}^{(i)}$ thus summarizes which base kernels are present that act on dimension $i$ or if the dimension is ignored. The distance in this case is defined as

$$
\begin{aligned}
d(\mathcal{T}_1, \mathcal{T}_2) := & \alpha_1 \sum_{i=1}^{D} W_{\tilde{d}}(\omega_{1,\text{base}}^{(i)}, \omega_{2,\text{base}}^{(i)}) \\
& + \alpha_2 W_{\tilde{d}}(\omega_{1,\text{paths}}, \omega_{2,\text{paths}}) \\
& + \alpha_3 W_{\tilde{d}}(\omega_{1,\text{subtrees}}, \omega_{2,\text{subtrees}}).
\end{aligned}
\tag{3}
$$

The reason for this distinction is that two kernels $k_1$ and $k_2$, which have different active dimensions, are considered to be farther apart with this distance, making it easier to allow variable selection, which is an important aspect of model selection.

Our defined function $d(\mathcal{T}_1, \mathcal{T}_2)$ induces indeed a pseudo-metric in the kernel-grammar generated kernel space, as shown in the following proposition. In particular, it fulfills the triangle inequality.

**Proposition 1** *Let $\tilde{\mathbb{K}}$ be the kernel space generated by a kernel grammar. Let $f : \tilde{\mathbb{K}} \mapsto \mathcal{T}(\tilde{\mathbb{K}})$ be a mapping that maps a kernel $k \in \tilde{\mathbb{K}}$ to one of its expression trees $\mathcal{T}$. Then $\hat{d}(k_1, k_2) := d(f(k_1), f(k_2))$ is a pseudo-metric on $\tilde{\mathbb{K}}$ where $d$ is given by (2) or (3) depending on the base kernels in $\tilde{\mathbb{K}}$.*

Given the pseudo-metric, we are now able to define a kernel-kernel with

$$K_{SOT}(k_1, k_2) := \sigma^2 \exp\left(\frac{-\hat{d}(k_1, k_2)}{l^2}\right), \tag{4}$$

which we call *Symbolical-Optimal-Transport (SOT)* kernel-kernel. Here, $l$ denotes the lengthscale and $\sigma^2$ the variance of the kernel-kernel. Both parameters are learned in combination with the distance weights via marginal likelihood maximization (see Appendix A).

### 3.1 Bayesian Optimization for Model Selection

We utilize the proposed kernel-kernel to do model selection for GP's via Bayesian optimization, which is a similar task to [9]. Compared to [9], we use our proposed kernel-kernel, which is a fundamentally different and computationally more efficient way of measuring similarity in GP space.

Table 1: RMSE on test kernel-log-evidence pairs ($\alpha = 0.05$)

| Dataset | Hellinger | kNN | Mean | SOT (ours) |
|---|---|---|---|---|
| Airfoil | 0.1773 (0.004) | 0.2231 (0.013) | 0.3769 (0.002) | **0.0936** (0.003) |
| Airline | 0.3569 (0.007) | 0.3813 (0.011) | 0.4013 (0.005) | **0.3464** (0.011) |
| LGBB | 0.3772 (0.021) | 0.6795 (0.025) | 0.8519 (0.016) | **0.2783** (0.008) |
| Powerplant | 0.1912 (0.012) | 0.2236 (0.011) | 0.2925 (0.009) | **0.0137** (0.005) |
| Concrete | 0.2489 (0.006) | 0.1806 (0.010) | 0.2912 (0.005) | **0.0451** (0.002) |

Given a model selection criteria $g_{\mathcal{D}} : \tilde{\mathbb{K}} \rightarrow \mathbb{R}$ for a given dataset $\mathcal{D}$, we want to solve $k^* = \arg\max_{k \in \tilde{\mathbb{K}}} g(k|\mathcal{D})$ via Bayesian optimization. We thus define a surrogate GP model for $g(k|\mathcal{D})$ via

$$f \sim \mathcal{GP}(\mu_c(\cdot), K_{SOT}(\cdot, \cdot)),$$

where $\mu_c(k) = c$ is the constant mean function. As BO acquisition function $a(k|\tilde{\mathcal{D}}_t)$ we use Expected-Improvement (EI). Here, $\tilde{\mathcal{D}}_t$ denotes the set of already queried kernel-selection-criteria pairs $(k, g(k|\mathcal{D}))$ for which the meta-GP posterior $f|\tilde{\mathcal{D}}_t$ is calculated. Starting with an initial set of pairs $\tilde{\mathcal{D}}_0$ we follow the standard BO iterations where in each iteration the acquisition function is maximized $k_t = \arg\max_{k \in \tilde{\mathbb{K}}} a(k|\tilde{\mathcal{D}}_t)$ given the current set of already evaluated kernel-selection-criteria pairs $\tilde{\mathcal{D}}_t$. Then the selection criteria is queried at the chosen kernel $g(k_t|\mathcal{D})$. A complete description of the method can be found in Algorithm 1 in Appendix A.

A crucial part in the BO cycle is the optimization of the acquisition function $\max_{k \in \tilde{\mathbb{K}}} a(k|\tilde{\mathcal{D}}_t)$. While for Euclidean spaces gradient-based methods or grid-based methods could be used to solve that task, this is not an option for a structured space like the considered grammar-generated kernel space. We propose using an evolutionary algorithm to optimize the acquisition function, which can be seen in Algorithm 2 in Appendix A. While an evolutionary algorithm seems to be a computationally intense procedure that takes place in each BO iteration, we emphasize that the evaluation of the acquisition function is very efficient for our proposed method. The reason for the efficiency is that the evaluations of our kernel-kernel $K_{SOT}(k_1, k_2)$ at two kernels $k_1, k_2 \in \tilde{\mathbb{K}}$ is very cheap, compared for example to the method in [9] (see Appendix E for computational time comparision).

### 3.2 Comparision to Hellinger Kernel-Kernel

The work of [9] also uses BO for kernel selection but with a different principle of measuring the distance in GP space. In particular, they propose measuring the distance of two GP's $\mathcal{M}$ and $\mathcal{M}'$ via the induced prior distributions $p(\mathbf{f}|\mathbf{X}, \mathcal{M})$ in function space evaluated on the design matrix $\mathbf{X}$ of the dataset. Conditioned on the parameters of the GP $\theta$, the distributions $p(\mathbf{f}|\mathbf{X}, \mathcal{M}, \theta)$ are Gaussian and they use the Hellinger distance over Gaussians as base distance between GP models $d(\mathcal{M}_\theta, \mathcal{M}'_{\theta'}|\theta, \theta')$ given the hyperparameters. They finally take the expectation over the hyperparameter priors to construct their final distance. Thus, for each kernel-kernel evaluation $K(\mathcal{M}, \mathcal{M}')$ the integral

$$d(\mathcal{M}, \mathcal{M}') = \int \int d(\mathcal{M}_\theta, \mathcal{M}'_{\theta'}|\theta, \theta')p(\theta)p(\theta')d\theta d\theta'$$

needs to be computed, where the integrand scales cubically in $|\mathbf{X}|$. Each kernel-kernel evaluation is therefore a computationally hard problem by itself. They side-pass this issue partly by selecting a subset of $\mathbf{X}$ as input locations and by using sample-based estimation of the integral. While this could make the method useful for oracles with very long run times, such as model selection for very large data sets, we see conceptual problems in the case of medium oracle run times. For our method, kernel-kernel evaluations $K(k_1, k_2)$ are very efficient as they scale only in the size of the expression tree and no integrals need to be computed.

## 4 Experiments

In the following section, we show experimental results for our novel meta-GP model and kernel search method. We evaluate our meta-model on a meta-regression task where we predict test kernel-log-evidence pairs based on training pairs. Secondly, we consider kernel search and compare it

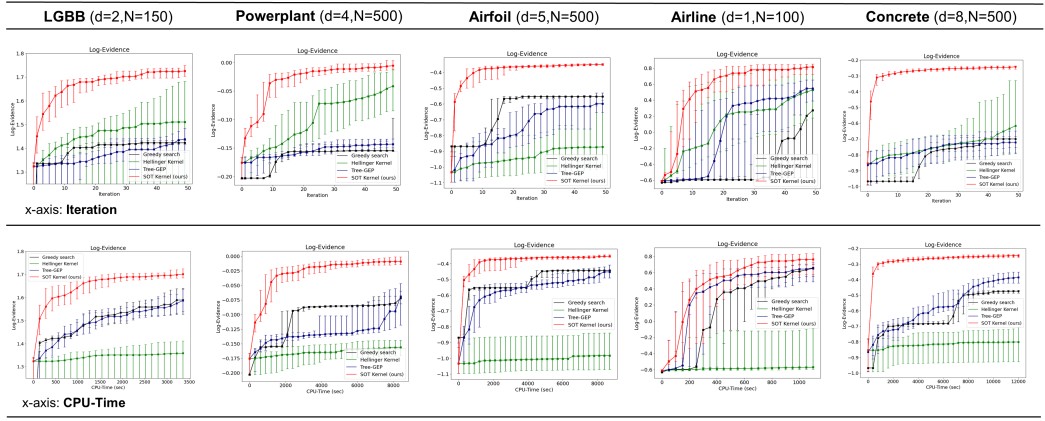

Figure 1: Plots for the model selection task over number of evaluated models (top) and CPU-time (bottom) showing the normalized log-evidence of the best model found up to this iteration/time point.

against the greedy method in [3], the evolutionary algorithm in [4] and against the BO method of [9]. The implementation of our method is available at `https://github.com/boschresearch/bosot`.

**Selection Criteria and Datasets:** In all our experiments, we consider the normalized log-model evidence $g(k|\mathcal{D}) = \log p(\mathbf{y}|\mathbf{X}, k)/|\mathcal{D}|$ as model selection criteria, as also done in [9]. We always use the Laplace approximation to calculate the log-model evidence, where we use 10 repeats to do MAP estimation of the kernel parameters. Furthermore, we consider the following publicly available datasets: Airline, LGBB, Airfoil, Powerplant, Concrete. Airline is a one dimensional time-series dataset, LGBB is a two-dimensional dataset with low observation noise. Powerplant, Airfoil and Concrete are four, five, and eight dimensional, whereas Powerplant exhibits higher observation noise. We use medium-sized training sets - 100 and 150 datapoints for Airline and LGBB and 500 datapoints for the other three datasets. All training sets are uniformly drawn from the full datasets. The outputs are normalized and the inputs are scaled to be in the unit interval. Further details can be found in Appendix D.

**Search Spaces:** We consider two search spaces. The first consists of the base kernels $\mathrm{SE}_i, \mathrm{LIN}_i, \mathrm{PER}_i, \mathrm{RQ}_i$ and the operators $+$ and $\times$. This is the space considered in [3] for time series and low-dimensional datasets. The second search space uses as base kernels $\mathrm{SE}_i$ and $\mathrm{RQ}_i$ and also $+$ and $\times$ as operators and was considered in [3] for higher dimensional base datasets. We consider the first space for Airline and LGBB and the second for Powerplant, Airfoil, and Concrete. The hyperparameter priors for the base kernels can be found in the Appendix A.

**Prediction of Selection Criteria:** We evaluate our meta-GP model on a meta-regression task by predicting test kernel-log-evidence pairs based on training pairs. The quality of predictions in kernel space might also be an indicator for good performance in BO for kernel search. Given the base dataset $\mathcal{D}$ we create a training set $\tilde{\mathcal{D}}_{\mathrm{train}} = \{(k_i, g(k_i|\mathcal{D})) | k_i \in \tilde{\mathbb{K}}_{\mathrm{train}} \subset \tilde{\mathbb{K}}, i = 1, \ldots, n_{\mathrm{train}}\}$ and a test set $\tilde{\mathcal{D}}_{\mathrm{test}} = \{(k_i, g(k_i|\mathcal{D})) | k_i \in \tilde{\mathbb{K}}_{\mathrm{test}} \subset \tilde{\mathbb{K}}, i = 1, \ldots, n_{\mathrm{test}}\}$ - each containing 500 kernel-log-evidence pairs. We generate the train and tests sets exactly as in [9], where we create one set $\tilde{\mathbb{K}}_{\mathrm{complete}}$ by first initializing it with all base kernels and iteratively pick one kernel of the current set, apply one random operation of the kernel grammar and add the resulting kernel to the current set and repeat this process until we have $n_{\mathrm{train}} + n_{\mathrm{test}}$ kernels in $\tilde{\mathbb{K}}_{\mathrm{complete}}$. We then divide the set uniformly into $\tilde{\mathbb{K}}_{\mathrm{train}}$ and $\tilde{\mathbb{K}}_{\mathrm{test}}$. We compare our model to the mean-predictor as baseline, which just predicts the mean of the train set at each test point and a kNN predictor based on the kernel-grammar operations [see Appendix D or [9]]. Furthermore, we compare against the meta-GP model of [9]. We report root mean squared error (RMSE) scores on the test sets in Table 1. It can be observed that our method leads to more precise predictions on all four meta prediction tasks, indicating that the symbolical representations already contain sufficient information to predict log-evidence values.

Table 2: Predictive negative log-likelihood values on test set at final time stamp. Values are marked bold if they are not significantly different from the best value according to a two-sample t-test ($\alpha = 0.05$).

| Dataset | Greedy | Hellinger | Tree-GEP | SOT (ours) |
|---------|--------|-----------|----------|------------|
| Airline | -0.4042 (0.615) | 0.3368 (0.207) | **-0.5594** (0.580) | **-0.7015** (0.471) |
| LGBB | -0.7528 (0.661) | **-0.8787** (0.854) | **-1.0701** (0.532) | **-0.9325** (0.492) |
| Powerplant | -0.0053 (0.054) | 0.0580 (0.037) | -0.0241 (0.057) | **-0.0661** (0.032) |
| Airfoil | **0.0837** (0.026) | 0.9080 (0.205) | 0.1826 (0.118) | **0.1006** (0.090) |
| Concrete | 0.3254 (0.019) | 0.6633 (0.207) | **0.2812** (0.074) | **0.2872** (0.044) |

**Model Selection - Setup:**   Concerning kernel search, we compare against the BO method that employs the Hellinger kernel-kernel [9], against greedy search [3] and against the evolutionary algorithm presented in [4], referred to as TreeGEP. Both BO methods run for 50 iterations and the kernel-kernel hyperparameters are updated in each iteration via marginal likelihood maximization. Our method applied the evolutionary Algorithm 2 (see Appendix A) to optimize its acquisition function, using a population size of 100. As it is computationally unfeasible to apply the same kind of acquisition function optimization for the Hellinger kernel-kernel we use their method of optimizing the acquisition function where an active set of kernels is updated in each iteration. Both BO methods and TreeGEP start with the same set of initial kernels, for which we apply two random grammar operations for each base kernel. Greedy search by design needs to start from the empty kernel. We give it a head start by the number of initial datapoints (see Appendix D). For each dataset, we display results from 30 independent runs with different seeds, namely medians and quartiles of the log-evidence score over iterations and CPU-time in Figure 1. The implementation of both BO methods is based on *GPflow* [10]. Further experimental details and parameter settings for all methods can be found in Appendix D.

**Model Selection - Results:**   As shown in Figure 1, we outperform all methods in terms of performance over number of model evaluations. This is not surprising against the two heuristics, as they are not optimized towards keeping the number of model evaluations low. However, it is notable that we are more sample-efficient compared to [9], who solve a much harder problem for computing their kernel-kernel. In terms of performance over CPU-time, we outperform all methods on Airfoil, Powerplant, LGBB and Concrete significantly. On Airline, the advantage over the heuristics is smaller - the reason is the lower oracle time - which benefits the heuristics that do not need to optimize the acquisition function. The reason for the poor performance of the Hellinger kernel-kernel in terms of CPU-time is the high ratio of acquisition function optimization to oracle time, which was as high as $50 : 1$ in our experiments (see detailed numbers in Appendix D).

**Test Performance:**   When optimizing a model selection criteria, one expects that this also materializes in a better test performance. In Table 2 we therefore show the predictive negative log-likelihood (NLL) scores on a held out test-set of the selected models at the final time stamp. We observe that the advantage in the model selection value gets transferred to the test performance.

**Further Experiments:**   In Appendix E we include further investigations on the behavior of our method on simulated data coming from a ground-truth kernel. Furthermore, we include a comparison of the selected kernel to the standard RBF kernel and against Functional Kernel Learning (FKL) [2].

## 5   Limitations

In case the dataset size is very small, greedy search or evolutionary algorithms might have an advantage in terms of computational time as the optimization of the acquisition function outweighs the computation of the model selection criteria. Thus, in these instances our method might not show a strong benefit. Furthermore, the kernel grammar is often used for downstream applications that use the selected hypothesis for interpretation, such as [7] who build an automatic natural language description of the selected hypothesis. Depending on the dataset and how many steps are employed in the acquisition function optimization, relatively large hypotheses might be found as optimal. This

might render the interpretation difficult. However, we show in Appendix E example configurations of our algorithm that can be used to get smaller, well interpretable hypothesis.

## 6 Conclusion

We presented a novel way of doing BO for model selection of GP's by measuring the distance between two GP's via the symbolical description of the underlying statistical hypothesis. The main contribution is the deduced pseudo-metric over kernels and the resulting SOT kernel-kernel. We show that our approach leads to a more efficient way of searching through a discrete kernel space compared to other BO methods and search heuristics.

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
