# OpenReview forum: "Structural Kernel Search via Bayesian Optimization and Symbolical Optimal Transport"
_NeurIPS.cc/2022/Conference — NeurIPS 2022 Accept_

### Official Review · Reviewer_iFD5 · 2022-07-01

**Rating:** 7
**Confidence:** 3
**Soundness:** 4 excellent
**Presentation:** 3 good
**Contribution:** 3 good

**Summary:**

This paper proposes a new way to select a kernel (covariance function) for Gaussian processes (GP). The authors borrow ideas from NN architecture search to propose a so called symbolical-optimal-transport (SOT) kernel over the architecture of the kernel. In contrast to prior work, they do not compare to different kernels in function space. Instead they compare the symbolic architecture of the kernel directly. They do this by capturing the structure of the kernel in a tree and then compare these trees by comparing them by some kind of optimal-transport metric.
This SOT kernel is then used for Bayesian optimization for model selection. This is fairly standard. A comparison to a hellinger kernel-kernel is given. The SOT kernel is computational advantageous because no integrals have to be computed.
The authors then demonstrate in a series of four experiments that the SOT kernel outperforms Greedy, Hellinger, Tree-GEP and SOT. This is true for both the training and the test set.

**Questions:**

--- How does this work together with hyper parameter tuning of GP? Can I first run your method and then tune the hyper parameters as usual?
--- Can this be used to pick the kernel for RKHS regression or SVMs?
--- On page 4: Are Rule 3 and Rule 2 ('exchanging a base kernel with another kernel') really necessary? It seems to that they assuming the other rules would be enough...

Minor points:
--- The plural of GP is GPs, not GP's.
--- Please define g before it's used on page 3.

**Limitations:**

The limitations are transparent and well-addressed.

**Strengths And Weaknesses:**

Strengths: While this idea has been executed for Neural Networks before, the presented idea is novel for Gaussian process model selection. The experiments are very encouraging and the idea seems natural. The computational advantages of this approach (over the known function space view) are well-explained. While the idea is simple, I do not consider this a weakness. The topic is very significant because model selection is extremely important for GPs. The quality of writing is good, it is quite easy to understand.

Weakness: There is almost no theory in the paper, except for the trivial Proposition 1. I think the focus on "Optimal Transport" is misleading because the special case of the OT distance that is used is quite trivial, and looks more like an l1 norm of the weights. Nowhere else is optimal transport used.

---

> ### Author Response · Authors · 2022-08-02
> **Comments to Reviewer iFD5**
>
> We thank Reviewer iFD5 for the positive and constructive feedback and will discuss the questions and comments below:
>
> *I think the focus on "Optimal Transport" is misleading because the special case of the OT distance that is used is quite trivial, and looks more like an l1 norm of the weights. Nowhere else is optimal transport used.*
>
> We agree with the reviewer that the focus of our work is not explicitly on the optimal transport techniques, but rather on showing that the kernel-kernel should be defined on the grammar expressions rather than directly on the function space distribution of the GP. We note that we also don't emphasize a focus on OT techniques in the abstract. We will also make this clearer in the introduction of the camera-ready version.
>
> The OT formulation offers one possible way of defining such a kernel over the grammar expressions. One advantage of that formulation, though, is that it allows in principle to incorporate other ground metrics, such as tree metrics used for neural architecture search (NAS) in Nguyen et al. (2021). This increases the flexibility of our proposed method/ our formulation.
>
> *How does this work together with hyper parameter tuning of GP? Can I first run your method and then tune the hyper parameters as usual?*
>
> see comments to all reviewers
>
> *Can this be used to pick the kernel for RKHS regression or SVMs?*
>
> We did not test this, but in principle yes, as our kernel-kernel is defined over the structural form of the kernel and doesn't use other parts tied to GPs.
>
> Similar to the presented GP selection, one might need to do an optimization of the kernel hyperparameters before calculating the model selection criteria. Applying our method to an SVM classifier might for example use our algorithm in an outer kernel selection loop, while the kernel parameters are optimized in an inner loop, e.g. also via BO. The accuracy on a validation set could be used as selection criteria in both loops.
>
> *Are Rule 3 and Rule 2 ('exchanging a base kernel with another kernel') really necessary? It seems to that they assuming the other rules would be enough...*
>
> We thank the reviewer for that question/remark. In fact, the kernel space would stay the same without rule 3 (or rule 2 for the general grammar). However, these rules are not only used to define the kernel space but also specify *directions* in the kernel space. These operations/ search directions are used in both BO methods (ours and in the Hellinger kernel approach) in the optimization of the acquisition function. Furthermore, the operations are used in Greedy Search to define the next stage of kernels and in TreeGEP to generate the next population of kernels.

---

> > ### Comment · Reviewer_iFD5 · 2022-08-03
> > **Thank you for the clarifications!**
> >
> > Thank you for these convincing clarifications! I will accordingly raise my score by one.

---

### Official Review · Reviewer_PF7c · 2022-07-07

**Rating:** 7
**Confidence:** 3
**Soundness:** 3 good
**Presentation:** 2 fair
**Contribution:** 3 good

**Summary:**

The paper proposes a method for kernel selection (search) wherein kernels are built from base kernels using fundamental operations (eg addition, multiplication etc).  Kernels so constructed are represented as trees, allowing their similarity to be evaluated based on their grammar (tree structure and assumptions regarding basic operations, eg their commutativity or otherwise) rather than e.g. distance between GPs.

**Questions:**

1. When constructing base kernels, do you "fix" hyperparameters like length-scales (for the SE kernels) for each base, or do you tune such parameters using e.g. max-likelihood?  Further:
1a. if the former, do you attempt to give some flexibility by e.g. having multiple SE kernels with different lengthscales as distinct base kernels?
1b. if the latter, how do you measure the similarity between two examples of the "same" base kernels with different lengthscales?

2. Similarly, when doing operations like ADD, do you considered allowing for weighted sums (a1.K1(x,y) + a2.K2(x,y), where a1,a2 are scaling factors) rather than simple sums (K1(x,y)+K2(x,y))?  This would allow more flexibility, but I'm not sure how well it would fit in your kernel similarity calculations.

3. As a rough order-of-magnitude, how complicated do SOT kernels become?  I would imagine this would depend on how long you let the BO run and what heuristic limits you place on the your evolutionary algorithm, but I'm curious how complex the tree becomes.

4. Does using an evolutionary algorithm rather than global optimisation when maximising your BO acquisition function adversely affect convergence guarantees that usually apply to BO?

**Limitations:**

As noted previously, more experimental comparisons (more datasets) and at least one non-parametric result (e.g. hyperkernels) would improve this paper.

**Strengths And Weaknesses:**

The construction of the kernel-kernel is thorough and I am persuaded that, while heuristic, this is a good alternative to measuring the difference between GPs or kernels using e.g. L2-norm.  Further, the overall approach of building kernels from base kernels reflects the human (parametric) approach to the problem well.

My main problems with this paper are:

1. Comparison with non-parametric approaches to kernel selection (such as hyperkernels) is missing.
2. It could be argued that this approach is incremental, simply substituting one kernel-kernel [2] with another that has the advantage of being more readily evaluated.
3. Experimental results are limited.

I would be more comfortable in overlooking the second point if there was more experimental results (ie. more than 4 datasets evaluated), but it is difficult to draw strong conclusions from limited results, even though the improvements shown appear strong.  However I think it is important to include comparisons with non-parametric approaches for completeness.

---

> ### Author Response · Authors · 2022-08-02
> **Comments to Reviewer PF7c**
>
> We thank Reviewer PF7c for the positive and constructive feedback and will discuss the questions and comments below:
>
> *I would be more comfortable in overlooking the second point if there was more experimental results (ie. more than 4 datasets evaluated), but it is difficult to draw strong conclusions from limited results, even though the improvements shown appear strong. However I think it is important to include comparisons with non-parametric approaches for completeness.*
>
> In order to address this point we added more experimental results to the revised version of the supplementary material in a new file called additional_experiments_rebuttal.pdf. We made a comparison to the most recent nonparametric kernel learning method we could find, called "Functional Kernel Learning" of Benton et al. (2019), which places a GP prior over the spectral density of kernels. We observe that almost all search methods over the kernel grammar (including our proposed one) lead to better performing models in the end. Furthermore, we added experiments for the UCI dataset "Concrete", which also can be found in Appendix E (experiments of FKL on Concrete were not yet finished and will be added later).
>
> *When constructing base kernels, do you "fix" hyperparameters like length-scales (for the SE kernels) for each base, or do you tune such parameters using e.g. max-likelihood?*
>
> see comments to all reviewers
>
> *1b. if the latter, how do you measure the similarity between two examples of the "same" base kernels with different lengthscales?*
>
> We thank the reviewer for mentioning this important point.
>
> We treat each expression and also the base kernels as *kernel-families over their parameters*. Comparing two identical families thus results in a distance of 0. We mention this in more detail in Appendix C and in footnote 1 in the main paper.
>
> For example, the symbol SE stands for the whole family of SE kernels over their lengthscale and variance parameters. The distance between two SE kernels thus is 0 as we deal with the same element, the same kernel-family. All other considered search methods also act on the kernel-family level. The whole procedure might be seen as two model selection loops. Our method searches over kernel families and acts as the outer selection loop. The inner selection loop selects the best parameters for a given kernel family - this is done automatically when calculating the model selection criteria (see comment at the top). We will make this point clearer in the camera-ready version of the main paper.
>
> *Similarly, when doing operations like ADD, do you considered allowing for weighted sums (a1.K1(x,y) + a2.K2(x,y), where a1,a2 are scaling factors) rather than simple sums (K1(x,y)+K2(x,y))?*
>
> All base kernels come with a variance term, as defined in the kernel grammar in Duvenaud et al. (2013). This automatically leads to a scaling of the base kernels. The similarity is measured on the kernel-family level (as stated above) which includes the scaling parameters.
>
> Thus, two expressions that both have the form $k_1=SE+SE$ and $k_2=SE+SE$ will both consider the same kernel family spanned by $K(x,y|\theta_1)+K(x,y|\theta_2)$ over the combined parameter space $\theta_1\times \theta_2$, which includes both lengthscales and variances. As $k_1$ and $k_2$ define the same kernel-family, they have distance 0. In case a prior is given on the kernel parameters (and thus also on the scaling parameters) this can also be seen as a distance between two Bayesian models M1 and M2, which have the same prior in function space.
>
> *As a rough order-of-magnitude, how complicated do SOT kernels become? I would imagine this would depend on how long you let the BO run and what heuristic limits you place on the your evolutionary algorithm, but I'm curious how complex the tree becomes.*
>
> We thank the reviewer for mentioning this.
>
>  The complexity of the best kernel found indeed depends mainly on three things: i) the limits of the evolutionary algorithm in the acquisition optimization ii) the model selection criteria iii) the dataset. After inspecting multiple runs, we found expressions consisting out of 6 - 16 base kernels.
>
> *Does using an evolutionary algorithm rather than global optimisation when maximising your BO acquisition function adversely affect convergence guarantees that usually apply to BO?*
>
> In fact, results on convergence rates in BO require that a global optimum of the acquisition function has been found. It might be an interesting research question what happens if the acquisition function is not optimized perfectly. However, this might also be a general problem in BO as the acquisition function in standard BO can get highly non-convex, making it hard to find the global optimum.

---

> > ### Comment · Reviewer_PF7c · 2022-08-03
> > **Response to rebuttal**
> >
> > Thank you for the rebuttal.  This addresses my concerns so I have raised my recommendation to accept.
> >
> >
> > With regard to BO convergence, it is reasonably common practice to use a global optimiser like DIRect or similar to maximise the acquisition function, which has convergence guarantees regardless of non-convexity.  However I accept that this case requires an alternative approach, and it is certainly not the alone in that regard - for example in high-dimensional problems, even if global optimisers are used, early stopping may be required for practical reasons.  While it is outside the scope of this paper it would certainly be interesting to see an analysis of the influence (or possibly lack thereof) that failure to optimise the acquisition function has on convergence guarantees in BO!

---

### Official Review · Reviewer_Cka7 · 2022-07-11

**Rating:** 7
**Confidence:** 4
**Soundness:** 4 excellent
**Presentation:** 4 excellent
**Contribution:** 3 good

**Summary:**

To find the best kernel for a GP, rather than conducting BO with a value function given by a GP comparison in function space, this paper develops a novel way to compare two kernels, based on optimal transport between feature representations of the tree representation of the kernel.

**Questions:**

Given that your method eschews functional distance like in [2], are we sure that it would do well in cases where the true kernel is known? I suppose this can be evaluated on synthetic data. Or is there some weakness here, hidden in the simplicity (both theoretical and computational) of the pairwise evaluation?
How can the OT distance be interpreted? Is there something of interest happening with the multiplicative factors in the total distance eq3 once they get learnt?

**Limitations:**

Yes, though section 5 could be made more explicit.

**Strengths And Weaknesses:**

The paper comes up with a good idea for kernel search, inspired I believe from references [10,4] which conduct NAS by BO using OT over the computational graph. Thus the central idea is at hand (structure search by BO using OT over structure distance), though not immediate.
The paper is clearly written. Many details are left to the appendix, but this still seems adequate.
The experimental validation is adequate.
The formalization of the OT metric in sec 3 is justified thoroughly.
Fig 1 is not legible when printed on A4 format.

---

> ### Author Response · Authors · 2022-08-02
> **Comments to Reviewer Cka7**
>
> We thank Reviewer Cka7 for the positive and constructive feedback and will discuss the questions and comments below:
>
> *Given that your method eschews functional distance like in [2], are we sure that it would do well in cases where the true kernel is known? I suppose this can be evaluated on synthetic data.*
>
> We thank the reviewer for the interesting question. In case synthetic data from a ground truth kernel $k_{gt}$ is used, one might consider two separate points:
>
> 1. Do we select the ground truth kernel in the end?
>
> 2. Do we reach the same log-evidence value as the ground-truth kernel aka $g(k_{gt}|D)$?
>
> The first point might depend on the model-selection criteria $g$ that is optimized with our method - e.g. which approximation is used for the log-evidence and if this approximation obtains its maximal value at the ground truth kernel. Considering the second point, we think that our method will select competitive kernels compared to $k_{gt}$ in terms of $g(k|D)$. We plan to add a small experimental section in the camera ready version considering this topic.
>
> Our expectation here is that if we would calculate the log-evidence perfectly and our dataset $D$ would be large enough than our search method might have a chance to select the ground truth kernel as the log-evidence would also probably have the highest value at the ground truth kernel. However, as we use a Laplace approximation for the log-evidence we observe that the model-selection criteria slightly prefers larger hypothesis/kernels (in terms of number of hyperparameters) over smaller ones - thus it might happen that a different kernel is selected, that has the same or even better model selection criteria value. However, our method is not tight to one special selection criteria. One could use for example BIC which tends to prefer smaller models, or one might use a computationally intense sampling approach to get a better estimate of the log-evidence.
>
> *Or is there some weakness here, hidden in the simplicity (both theoretical and computational) of the pairwise evaluation?*
>
> We did not observe that the simplicity of the kernel-kernel did harm the performance on the real-world datasets - we thus also don't expect that to happen for synthetic data.
>
> We think that the grammar expressions act like low-dimensional, compressed representations of the distributions in function space, which in turn allows the usage of simple distance measures - this might be the reason for the good performance despite the simplicity. One might compare this to SVM classification with a simple RBF kernel on image data, like MNIST, which can work fairly good in case well-chosen low-dimensional features of the high-dimensional image are used as input to the kernel - in our case these well-chosen features are already naturally given through the grammar expressions.
>
> *How can the OT distance be interpreted? Is there something of interest happening with the multiplicative factors in the total distance eq3 once they get learnt?*
>
> We thank the reviewer for mentioning this interesting topic.
>
> The distance value $d(k_{1},k_{2})$ itself might be hard to interpret. However, the multiplicative factors indeed reveal which parts of the distance are more important, depending on the base dataset for which the selection criteria $g(k|D)$ is calculated.
>
> This can be seen for example in Appendix E, Figure 4, where the values of the factors over the BO iterations for the Airfoil and Airline dataset are shown. Here, we observe that for the Airfoil dataset a comparison  of the base kernels (a high weight on the distance over base kernels) is more important while on the Airline dataset the distance over subtrees is more important - this might be an indication that for the Airline dataset very specific kernel-operator combinations are important while on Airfoil selecting the correct set of base kernels is more important.

---

### Official Review · Reviewer_i6Db · 2022-07-24

**Rating:** 7
**Confidence:** 3
**Soundness:** 3 good
**Presentation:** 3 good
**Contribution:** 3 good

**Summary:**

This paper considers the problem of selecting the kernel of a Gaussian process (i.e., the model selection problem), which is one of the hyperparameter optimization problems for Bayesian optimization.
The authors considered that a kernel function is consisted of a set of "basis" kernels in the space of kernel functions and a number of operations, following the previous (kernel grammer) work of Duvenaud et al.
A kernel grammar can be mapped to a tree that represents it.
For the discrete probability distribution that is a summary of this representation tree, we can define the Wasserstein distance with the indicator function as the cost function. In this case, the Wasserstein distance coincides with the total variation distance. This distance can be used to define a meta kernel function (kernel-kernel) that measures the similarity of two kernel functions on the space of kernel functions. The proposed method uses a Gaussian process with this kernel-kernel as the covariance function, and Bayesian optimization is used to select the kernel model.
In the experiment, the authors compared and evaluated the model selection performance of the proposed method with existing methods on four types of benchmark data including time series.

**Questions:**

- The proposed method seems to cover only model selection for the kernel function itself, but if we want to further perform hyperparameter selection for the kernel, can we combine it with conventional methods (e.g., marginal likelihood maximization)?

- Is it possible to approximate the Wasserstein distance and use the proposed method when a general distance that is not an indicator function is used as the base distance?

- Are the hyperparameters alpha_1, 2, and 3 in distance (3) optimized at the same time as the other kernel-kernel hyperparameters? How does this affect performance of the proposed method?

**Limitations:**

The authors adequately addressed the limitations and potential negative societal impact of their work.

**Strengths And Weaknesses:**

strength

- Traditionally, model selection for the kernel function itself has been done manually depending on the problem or by multiple kernel modeling for several candidate kernels. On the other hand, this study performs Bayesian optimization of combinations of kernel components based on the similarity defined between the kernel functions. In this approach, only the base kernel and operation rules need to be prepared, and there is no need to prepare multiple candidate kernel functions as in multiple kernel learning. Bayesian optimization for model selection allows for the "construction" of good kernel functions without human intervention.

- The mapping of kernels to representation trees allows the problem of finding the optimal kernel to be viewed as a problem of finding the optimal tree structure.

- The proposed method has higher scalability than existing methods (Malkomes et al.) that use Bayesian optimization based on Hellinger distance for the same problem. This is because the Hellinger distance-based method requires an integral calculation on the kernel parameters for kernel-kernel evaluation.

weakness

- In this paper, an indicator function is used for the base distance to describe the Wasserstein distance as a closed-form solution (total variation distance). However, this seems to be a discrete and extreme similarity evaluation that only looks at whether or not the elements that make up the two representation trees match.

- The distance for kernel-kernel (3) itself contains the hyperparameters alpha_1, 2, and 3, so it seems that the hyperparameter tuning for kernel-kernel are going to be harder than for ordinary BO when performing BO on the kernel space.

---

> ### Author Response · Authors · 2022-08-02
> **Comments to Reviewer i6Db**
>
> We thank Reviewer i6Db for the positive and constructive feedback and will discuss the questions and comments below:
>
> *In this paper, an indicator function is used for the base distance to describe the Wasserstein distance as a closed-form solution (total variation distance). However, this seems to be a discrete and extreme similarity evaluation that only looks at whether or not the elements that make up the two representation trees match*
>
> It is true that the indicator function is a very simple basic metric and in fact only considers feature match. However, this is compensated by the fact that the feature match is measured on a diverse set of features, namely the base kernels, paths, and subtrees, which also represent different levels of granularity.
>
> For example, the contribution of two subtrees $S_{1}$ and $S_{2}$ to the total distance between two kernels is determined not only by whether the subtrees completely match - but also whether their own subtrees match, or whether their base kernels match - this makes the overall distance expressive even though it has a simple base distance.
>
> *Is it possible to approximate the Wasserstein distance and use the proposed method when a general distance that is not an indicator function is used as the base distance?*
>
> Yes, in case one wants to use a different ground distance, there are two options:
>
> 1. One could compute the Wasserstein distance by solving the optimal transport problem using linear programming for each kernel evaluation - this was done for neural architecture search in Kandasamy et al. (2019) and for BO over molecules in Korovina et al. (2020). However, kernels based on general Wasserstein distances are not necessarily p.s.d.
>
> 2. One could consider tree metrics as a basic metric. These were used for neural architecture search in Nguyen et al. (2021) and allow for more flexibility. For these kinds of metrics, the OT problem also has closed form solutions and the resulting kernel is also p.s.d. We conducted some tests with these metrics and did not find any significant change in performance. Therefore, we decided to use the conceptually simpler ground metric.
>
> *The proposed method seems to cover only model selection for the kernel function itself, but if we want to further perform hyperparameter selection for the kernel, can we combine it with conventional methods (e.g., marginal likelihood maximization)?*
>
> see comments to all reviewers
>
> *Are the hyperparameters alpha1, 2, and 3 in distance (3) optimized at the same time as the other kernel-kernel hyperparameters? How does this affect performance of the proposed method?*
>
> Yes, all hyperparameters of the kernel-kernel, including the distance weights, lengthscale and variance are learned in a combined way via marginal-likelihood maximization using the LBFGS optimizer.
>
> We also think that this is the correct way, as we want to find the combined set of hyperparameters with highest marginal-likelihood value - thus solve for the combined optimization problem. Furthermore, we observed that the optimization of all hyperparameters combined is fairly stable and usually ends up at the same values for different starting points, indicating a well-behaved loss landscape for the combined optimization problem.
>
> *The distance for kernel-kernel (3) itself contains the hyperparameters alpha1, 2, and 3, so it seems that the hyperparameter tuning for kernel-kernel are going to be harder than for ordinary BO when performing BO on the kernel space.*
>
> Our hyperparameter optimization problem is comparable to standard BO with an RBF kernel on $\mathbb{R}^{d}$ containing the same number of hyperparameters. Since we learn five hyperparameters it is comparable to BO over four dimensions with an RBF kernel with four lengthscales and one variance.
>
> However, the hyperparameter optimization step is not necessarily the most computationally intensive part of GP inference in kernel space. For the Hellinger kernel, the distance calculations, for example, consume much more computational time than the hyperparameter optimization itself (which can be performed without recalculating the distances in each iteration). Our kernel-kernel on the other hand performs distance calculations much faster (see Appendix D, Figure 3) - rendering BO over kernel space much more similar (in terms of computation time) to standard BO over euclidean spaces.

---

> > ### Comment · Reviewer_i6Db · 2022-08-08
> > **Thank you for the careful responses.**
> >
> > Thank you for your careful response to each question.
> > I think it is great that positive results are obtained, especially for the validity of the choice of base distance and the hyperparameter optimization of the proposed method, which I wanted to know.
> > I continue to recommend the acceptance.

---

### Author Response · Authors · 2022-08-02
**Comment to all reviewers**

**Changes in the revised version:**

A new file named additional_experiments_rebuttal.pdf that addresses the experiment requests of reviewer PF7c is included in the supplementary zip. All other proposed changes and clarifications will be included in the camera-ready version of the paper/supplementary.

**Optimization of the kernel hyperparameters:**

We thank the reviewers for bringing to our attention that we were not clear enough on how we proceed with the GP/kernel hyperparameters of the selected kernels $k_{t}$ during the BO iterations. We actually *learn the hyperparameters* of the kernel as part of calculating the model selection criteria.

As we approximate the log-evidence $g(k_{t}|D)$ via Laplace approximation, the calculation of $g(k_{t}|D)$ already includes an MAP optimization of the kernel hyperparameters (see Appendix A.3) - we thus automatically receive learned hyperparameters when calculating the log-evidence. In fact, most of the GP model selection criteria, such as the utilized log-evidence, but also the Bayesian information criteria (BIC) implicitly perform a hyperparameter optimization. We will make this point clearer in the camera-ready version of the main paper.

---

> ### Public Comment · ~Carlos_Mondragon1 · 2024-04-05
> **<h1>html</h1>**
>
> <h1>html</h1>

---

### Meta-Review · Area_Chair_Yg8U · 2022-08-24

**Recommendation:** Accept
**Confidence:** Certain

**Metareview:**

This is a strong submission that benefitted greatly from productive and clarifying discussion between the authors and reviewers, after which the reviewers reached a unanimous stance in favor of acceptance. I recommend the authors to revise the manuscript accordingly in light of these discussions.

**Award:**

No

---

### Decision · Program_Chairs · 2022-09-14

Accept